# Maternal Dietary Inflammatory Index during Pregnancy Is Associated with Perinatal Outcomes: Results from the IMPACT BCN Trial

**DOI:** 10.3390/nu14112284

**Published:** 2022-05-29

**Authors:** Rosa Casas, Sara Castro-Barquero, Francesca Crovetto, Marta Larroya, Ana Maria Ruiz-León, Laura Segalés, Ayako Nakaki, Lina Youssef, Leticia Benitez, Francesc Casanovas-Garriga, Eduard Vieta, Fàtima Crispi, Eduard Gratacós, Ramon Estruch

**Affiliations:** 1Centro de Investigación Biomédica en Red de Fisiopatología de la Obesidad y Nutrición (CIBERON), Instituto de Salud Carlos III, 28007 Madrid, Spain; rcasas1@clinic.cat (R.C.); sacastro@clinic.cat (S.C.-B.); amruiz@clinic.cat (A.M.R.-L.); restruch@clinic.cat (R.E.); 2Department of Internal Medicine Hospital Clinic, Institut d’Investigacions Biomèdiques August Pi i Sunyer, University of Barcelona, 08028 Barcelona, Spain; frcasang7@alumnes.ub.edu; 3BCNatal|Fetal Medicine Research Center, Hospital Clínic and Hospital Sant Joan de Deéu, Institut d’Investigacions Biomèdiques August Pi i Sunyer (IDIBAPS), University of Barcelona, 08028 Barcelona, Spain; larroya@clinic.cat (M.L.); segaleslaura@gmail.com (L.S.); anakaki@clinic.cat (A.N.); lyoussef@clinic.cat (L.Y.); lbenitez@clinic.cat (L.B.); fcrispi@clinic.cat (F.C.); gratacos@clinic.cat (E.G.); 4Centre for Biomedical Research on Rare Diseases (CIBER-ER), 28029 Madrid, Spain; 5Department of Psychiatry and Psychology, Hospital Clinic, Neuroscience Institute, Institut d’Investigacions Biomèdiques August Pi i Sunyer, CIBERSAM, University of Barcelona, 08028 Barcelona, Spain; evieta@clinic.cat

**Keywords:** Mediterranean diet, Dietary Inflammatory Index, pregnancy, birthweight percentile

## Abstract

The information available on the effects of maternal dietary habits on systemic inflammation and adverse maternal outcomes is limited. We aimed to evaluate whether Dietary Inflammatory Index (DII) score during pregnancy is associated with maternal body mass index (BMI), Mediterranean diet (MD) adherence, and perinatal outcomes. At 19–23 weeks’ gestation, 1028 pregnant women were recruited. Dietary information was assessed using a 17-item dietary score to evaluate MD adherence and a validated 151-item food frequency questionnaire. DII score was established according to 33 food and nutritional proinflammatory and anti-inflammatory items. Participants were distributed into tertiles according to the DII score, where a lower DII score (first tertile) represented an anti-inflammatory diet and the third tertile represented the more proinflammatory diet. Maternal characteristics and perinatal outcomes were collected, and newborns’ birthweight percentiles were calculated. Adjusted logistic regression models were used to assess the association of the DII score with maternal and perinatal characteristics, setting the third tertile as the reference group. Women in the third tertile showed lower adherence to MD score compared to the first tertile: median (25th to 75th percentile) 9 (7 to 11) vs. 6 (4.25 to 8), *p* < 0.001. The proinflammatory diet was significantly associated with a higher maternal pre-pregnancy BMI (adjusted β = 0.88; 95% CI: 0.31 to 1.45) and lower newborn’s birthweight percentile (adjusted β = −9.84th; 95% CI: −19.6 to −0.12). These data show that a proinflammatory diet profile may be associated with maternal overweight and fetal undergrowth.

## 1. Introduction

Pregnancy involves several physiological changes to promote fetal growth and prepare the mother for delivery, including a systemic immunoinflammatory response [1]. Although gestation is considered to have an anti-inflammatory profile, some pregnancy stages are characterized by a proinflammatory status with high levels of proinflammatory cytokines, interleukins, and growth factors [2]. The balance between proinflammatory and anti-inflammatory molecules determines a physiological versus complicated pregnancy course [3]. A proinflammatory pregnancy environment has been associated with adverse pregnancy outcomes, such as miscarriage, idiopathic recurrent pregnancy loss, prematurity, diabetes, intrauterine infections, small for gestational age (SGA), fetal growth restriction, and preeclampsia [4,5,6,7].

Diet significantly contributes to increasing the pregnancy risk from early gestation [8], playing a key role in the regulation of chronic inflammation in pregnant and non-pregnant adults [9,10,11,12,13,14,15,16,17]. In fact, several high-quality studies, reviews, and meta-analyses have demonstrated that the Mediterranean diet (MD) is a healthy dietary pattern and that higher adherence to MD is associated with the prevention of several diseases and lower inflammatory levels [18,19,20,21]. The MD is characterized by high consumption of fruits, vegetables, whole grain cereals, legumes, fish, and nuts; low–moderate intake of dairy products and wine; and limited consumption of red meat and processed meat. From the nutritional point of view, the MD is low in saturated fat and high in antioxidants (vitamin E, vitamin C), fiber, and healthy fats (monounsaturated and polyunsaturated fats (MUFAs and PUFAs)) mainly derived from extra virgin olive oil (EVOO) and oily fish (n-3 PUFAs) [22].

In the last few years, the interest in evaluating the overall inflammatory effects of diet has led to the design of a new tool that is useful in assessing the potential anti- and proinflammatory effects of an individual’s diet through a continuous scale, namely the Dietary Inflammatory Index (DII). The DII is based on a comprehensive review of the published literature in which each dietary parameter is given a score based on its effects on six inflammatory biomarkers [23]. This index has been mainly used in studies on non-pregnant populations. As far as we know, only one study has investigated the association between maternal diet during pregnancy and pregnancy outcomes using the DII [24]. In the study by Sen et al. [24], 1808 mother–child pairs in a pre-birth cohort in Massachusetts (Project Viva) have been evaluated. The results demonstrated that a proinflammatory diet during pregnancy is associated with maternal systemic inflammation and impaired fetal growth [24]. Another study by de Andrade Miranda et al. has reported an association between a proinflammatory diet and inadequate birthweight including SGA and large for gestational age infants [25]. In addition, our group has demonstrated—for the first time—in a recent clinical trial (the Improving Mothers for a better PrenAtal Care Trial Barcelona (IMPACT BCN)) with more than 1200 pregnant women involved that a structured lifestyle intervention during pregnancy can reduce SGA for which no previous treatment gave positive effects [26]. Specifically, a nutritional intervention—based on MD—was applied in one of the trial arms and demonstrated a reduction in the incidence of SGA by 36% (14% in the MD group vs. 21.9% in the non-intervention group) and perinatal complications by 26% (18.6% in the MD group vs. 26% in the non-intervention group) [26].

Given the scarce and inconsistent scientific evidence on maternal proinflammatory diet in pregnant women at high risk for SGA, we decided to investigate the association between maternal DII and maternal pre-pregnancy body mass index (BMI) and newborn birthweight percentile in high-risk women who were included in the IMPACT BCN trial.

## 2. Materials and Methods

### 2.1. Study Design and Participants

This study represents a secondary analysis of the IMPACT BCN trial, a randomized clinical trial with a parallel group conducted at a University Hospital in Barcelona, Spain (2017–2020) including 1221 pregnant women at high risk for SGA randomly allocated at 19–23 weeks’ gestation into three groups: an MD intervention, a stress reduction program or non-intervention. Participants in the MD group (*n* = 407) received monthly individual and group educational sessions and free provision of extra-virgin olive oil and walnuts. Women in the stress reduction group (*n* = 407) underwent an 8-week mindfulness-based stress reduction program adapted for pregnancy, consisting of weekly 2.5 h sessions and one full-day session. Women in the non-intervention group (*n* = 407) received pregnancy care as per institutional protocols. The Institutional Review Board of the Hospital Clínic of Barcelona approved the study (HCB-2016-0830). All mothers provided written informed consent.

The main results of this trial have been previously described [26] and the study protocol has been described elsewhere [27]. For the present analysis, from the total sample of 1221 randomized participants, 215 participants were excluded because of missing data on dietary information, and 36 participants showed extreme energy intake outside predefined limits [28], including 970 participants at high risk of developing SGA during pregnancy in the final dataset (Figure 1).

### 2.2. Assessment of Dietary Intake

A 151-item semi-quantitative food frequency questionnaire (FFQ) validated for the present study population [29], 7-day dietary recalls of the previous 7 days before the meeting, and a 17-item MD adherence score were administered by trained dietitians in a face-to-face interview at trial enrollment (19–23 weeks). Food items were listed under 151 food groups: milk and dairy products, cereals and whole grains, vegetables, legumes, sausages, oils and fats, eggs, meat and fish, fast food, canned products, fruit, nuts, sweets and desserts and others (salt and sugar), and alcoholic and non-alcoholic beverages. Food consumption derived from the FFQ and food records was converted into energy and nutrient intake with the CESNID and Moreiras composition tables using traditional recipes [30,31]. More detailed FFQ validation can be found [29]. Participants filled out the 7-day dietary recalls. Detailed instructions about portion sizes and how to provide these household measures were also included in the food diary.

We asked all women about their dietary habits during pregnancy over the 3 months that preceded their enrollment in the study. The study participants were classified according to the 17-item MD adherence score as low (<6 points), medium (6–11 points), and high adherence (≥12 points). Participants indicated their usual and frequent consumption of listed food items in the FFQ, based on nine frequency categories (ranging from never or <1 time/month to ≥6 times/day) and using common units or portion sizes. A total of 14 food groups were listed: milk and dairy products, cereals and whole grains, vegetables, legumes, sausages, oils and fats, eggs, meat and fish, fast food, canned products, fruit, nuts, sweets and desserts and others (salt and sugar), and alcoholic and non-alcoholic beverages.

### 2.3. DII Assessment

DII score was calculated from the FFQ data for all participants using the methodology of Shivappa et al. [21]. To create the DII score, we first calculated the mean and standard deviation (SD) for each food parameter. In the present study, we included 33 food parameters to construct this score: 9 proinflammatory food parameters (energy, carbohydrate, fat, protein, cholesterol, saturated fat, trans-fat, vitamin B12, and iron intake) and 24 anti-inflammatory parameters (MUFAs, PUFAs, n-3 and n-6 fatty acids, fiber, vitamin B6, folic acid, niacin, riboflavin, thiamin, vitamin A, vitamin C, vitamin D, vitamin E, β-carotene, oregano, pepper, onion, garlic, tea, zinc, selenium, and magnesium intake). We calculated the z-score by subtracting the ‘‘standard global mean’’ from the amount reported by each participant and dividing this value by the “global standard deviation”. To reduce “right skewness”, this z-score was converted to a centered percentile score. Then, we multiplied this score by the respective food parameter effect score derived from the Shivappa et al. study [23]. Finally, we summed all 33 food DII scores to create the overall DII score for each participant.

### 2.4. Maternal Characteristics

Maternal characteristics were obtained from different questionnaires and interviews administered to study participants and included maternal age, ethnicity, socioeconomic status (low/medium/high), occupation status, educational level, pre-pregnancy BMI, chronic hypertension, diabetes, parity (multiparous/nulliparous), adverse obstetrical history (fetal growth restriction, preeclampsia, stillbirth), use of assisted reproductive technologies, smoking during pregnancy, alcohol habits during pregnancy, yoga/relaxation during pregnancy, exercise during pregnancy, and baseline MD score (low/medium/high). Obesity was defined if pre-pregnancy BMI was above 30 kg/m^2^.

### 2.5. Perinatal Outcomes

Gestational diabetes mellitus (GDM): The screening and diagnosis of GDM were performed during the second trimester of gestation (24–28 weeks). The two-step approach to testing for GDM was based on first screening with the administration of a 50 g oral glucose solution followed by a 1 h venous glucose determination. Women whose glucose levels met or exceeded 140 mg/dL (7.8 mmol/L) underwent a 100 g, 3 h diagnostic oral glucose tolerance test (OGTT). GDM was diagnosed in women who had two or more abnormal values on the 3 h OGTT. Reference values for the 3 h OGTT were: basal glucose < 105 mg/dL (5.8 mmol/L), 1 h glucose < 190 mg/dL (10.6 mmol/L), 2 h glucose < 165 mg/dL (9.2 mmol/L), and 3 h glucose < 145 mg/dL (8.1 mmol/L).

Preeclampsia was defined as systolic blood pressure (SBP) ≥ 140 mmHg or diastolic blood pressure (DBP) ≥ 90 mmHg at least 4 h apart after 20 weeks of gestation and proteinuria of ≥300 mg in 24 h, according to specific guidelines [32].

Newborns’ birthweight: Birthweight percentile was calculated with birthweight adjusted by gestational age at delivery and gender, according to standards for the Spanish population [26]. SGA was defined as birthweight below the 10th percentile, severe SGA as birthweight below the 3rd percentile [33,34].

Preterm birth was defined as delivery < 37 weeks’ gestation [35].

Adverse perinatal outcome (APO) was defined as a composite score of preterm birth, preeclampsia, perinatal mortality, severe SGA, neonatal acidosis, and Apgar score below 7 at 5 min, or the presence of any major neonatal morbidity.

### 2.6. Statistical Analysis

First, participants were distributed into tertiles according to DII score. Descriptive statistics with the mean ± SD for the participants’ baseline characteristics were applied. Categorical variables are expressed as percentages. The comparison between DII score tertiles involved the use of one-way analysis of variance (ANOVA) for continuous variables and chi-square test for categorical variables, with tertile 1 as the lowest DII score and tertile 3 as the highest DII score. Additionally, ANOVA was used to determine differences in the baseline dietary intakes of nutrients and food parameters among the three DII score tertiles. Normality was assessed for all the variables using the Kolmogorov–Smirnov test. For non-normally distributed variables, differences between the groups were assessed using the Kruskal–Wallis test. For non-normally distributed variables, the values are expressed as median (25th, 75th percentile). Logistic regression models were used to assess the association of DII score with different clinical pregnancy outcomes: GDM, preeclampsia, SGA, severe SGA, and preterm birth. Tertile 3 (more proinflammatory diet) was set as the reference group. Data were expressed using odds ratios and their corresponding 95% confidence intervals. Model 1 was unadjusted, Model 2 was adjusted for age at enrollment, pre-pregnancy body weight, socioeconomic status (low/medium/high), ethnicity/race (Asian/Black/Latin American, White/others), number of cigarettes smoked during this period, alcohol consumption during the first trimester (yes/no), parity (nullipara/multipara), energy intake (kcal/day), intervention arm (control/stress reduction/MD), yoga and pilates practice (hours per week), weight gain during pregnancy, and assisted reproductive technologies (yes/no).

Finally, linear regression analyses were used to investigate the possible associations of maternal DII score with pre-pregnancy BMI and newborn’s birthweight percentile. In this case, Tertile 1 (more anti-inflammatory diet) was set as the reference group. Four models were constructed. Model 1: unadjusted; Model 2: adjusted for intervention arm (control/stress reduction/MD), body weight, and gestational age at enrollment; Model 3: model 2 + socioeconomic status (low/medium/high), yoga and pilates practice (hours per week), and assisted reproduction techniques (yes/no); Model 4: model 3 + age, parity (nullipara/multipara), ethnicity/race (Asian/Black/Latin American, White/others), alcohol and smoking habits (yes/no). Data were shown as β (95% CI).

We used SPSS software (SPSS Inc., version 22, IBM, Chicago, IL, USA) for all statistical analyses. *p* value < 0.05 was considered statistically significant.

## 3. Results

### 3.1. Characteristics of the Study Population According to DII Tertile

Table 1 shows the comparison of maternal and perinatal characteristics according to DII tertiles. Among 970 included participants, the mean maternal age was 37 ± 4.7 years and pre-pregnancy BMI was 23.9 ± 4.8 kg/m^2^. Most women were of White ethnicity (≥80%), with a socioeconomic class defined as high (63.7%). Regarding medical history, 11.2% of participants were obese, 4.9% had diabetes, and 4.2% had chronic hypertension. The DII mean in this study was −2.94 ± 1.12 units. Significant differences were observed for maternal pre-pregnancy BMI and exercise during pregnancy (*p*-value 0.011 and <0.001, respectively). No significant differences among DII tertiles were observed in socioeconomic status; age; or perinatal outcomes including GDM, preeclampsia, birthweight, cesarean section, gestational age at delivery, birthweight percentile, SGA, severe SGA, prematurity, and APO.

### 3.2. Maternal Dietary Characteristics and Adherence to MD According to DII Tertile

Overall, when comparing participants according to DII score, participants with the highest score (proinflammatory) showed lower consumption of nuts, fruits, vegetables, legumes, fish and seafood, lean meat, dairy products, onion, garlic, pepper, oregano, EVOO, and alcohol and higher intake of refined cereals, and processed meat. Lower adherence to MD was also observed in the MD score median (25th to 75th percentile) 9 (7 to 11) vs. 6 (4.25 to 8), *p* < 0.001. Data are shown in Table 2.

Table 3 shows the dietary intake of participants among DII tertiles. Women in the tertile 3 (proinflammatory) consumed lower levels of energy, protein, dietary fiber, total fat, saturated fatty acids (SFAs), MUFAs, PUFAs, linoleic and α-linolenic acid, eicosapentaenoic acid (EPA), docosahexaenoic acid (DHA), and dietary cholesterol, as well as lower consumption of micronutrients (vitamins and minerals). Tertile 3 showed a higher intake of trans fat compared to other tertiles.

### 3.3. Association of DII with Maternal BMI and Newborn’s Birthweight

Regarding the pre-pregnancy BMI, significant associations were observed in Tertile 3 and pre-pregnancy BMI (full-adjusted β = 0.88 kg/m^2^; 95% CI: 0.31 to 1.45) (Table 4). Maternal DII score (as a continuous variable) was directly associated with pre-pregnancy BMI (full-adjusted β = 0.32 kg/m^2^ per 1-unit increase in DII; 95% CI: 0.12 to 0.52).

For women allocated in the highest tertile of DII score (Table 4), we found significant associations between DII score and low birthweight percentile, while no significant associations were observed for tertile 2 (full-adjusted β = −9.84th; 95% CI: −19.57 to −0.12). Maternal DII score (as a continuous variable) was inversely associated with newborn birthweight percentile (full-adjusted β = −4.05th per 1-unit increase in DII; 95% CI: −7.42 to −0.68).

## 4. Discussion

The present study shows a direct association between an anti-inflammatory DII and lower pre-pregnancy BMI and higher newborn birthweight percentile. Moreover, individuals with higher DII present higher adherence to the adapted 17-point MD adherence score and healthier nutritional profile. To our knowledge, this is the first study that assessed the relationship between DII and maternal BMI and birthweight percentile in pregnant women at high risk.

### 4.1. Nutritional Intake According to DII Index

Our results showed that individuals in the lowest DII tertile (mostly anti-inflammatory diet) had a higher intake of antioxidants; vitamins and minerals; fiber; and PUFAs, including omega-3 derivatives. Both antioxidants present in fruits and vegetables and dietary fiber have been negatively associated with inflammation, which contributes to less lipid oxidation [36]. Although higher protein intake was associated with a higher DII score, lower protein intake was observed with a higher DII score, which was consistent with other studies [24,37,38]. However, depending on the protein food sources, protein intake may have different effects on the inflammatory response, i.e., plant-based vs. animal protein food sources [39,40].

As for food groups, individuals in the lowest DII tertile consumed more plant-based products, such as EVOO, nuts, vegetables, and fruits, and healthy animal protein food sources such as blue fish, and these findings are consistent with previous studies [41,42,43].

DII score has been postulated as a potential dietary advice tool as guidance for individuals in setting dietary goals to reduce inflammatory levels associated with unhealthy dietary patterns [23].

### 4.2. Mediterranean Diet Adherence

Participants with higher MD adherence showed lower DII. The anti-inflammatory and immunomodulating effects of MD are well known and encompass downregulating the expression of leukocyte adhesion molecules and decreasing proinflammatory molecules such as interleukins, chemokines, or soluble endothelial adhesion molecules, among others [44]. In this sense, Kibret et al. [45] observed a significant association between adherence to a healthy dietary pattern and lower GDM risk (OR 0.78 (95% CI 0.56 to 0.99)).

Another potential mechanism underlying health benefits associated with healthy dietary patterns is the modulation of gut microbiota, which has been linked to metabolic dysfunction during pregnancy [46].

Several studies showed that high adherence to healthy dietary patterns, such as MD, was associated with a lower DII score [37,43,47]. Both dietary quality scores share common dietary components, which can explain why both scores showed potential health benefits during pregnancy.

### 4.3. Perinatal Outcomes and Birthweight

We did not find significant associations between DII score and SGA. However, significant inverse associations were observed between DII and newborn birthweight percentile. Similar findings were described in Project Viva, a longitudinal cohort of 2128 mother–child pairs from Massachusetts, USA [24]. Our results support the evidence, suggesting that a maternal diet rich in antioxidants (e.g., fruits and vegetables) and a low DII score drive a newborn weight appropriate for gestational age at birth [48,49].

The links between pre-pregnancy BMI and adverse pregnancy outcomes such as preeclampsia [50], GDM, and preterm delivery [51] may be mediated by inflammatory status [52]. However, the mechanisms involved between pre-pregnancy BMI and inflammation remain unclear. It must be noted that pre-pregnancy BMI and inflammation are both linked to dietary patterns. Obesity during pregnancy is associated with many obstetric and perinatal complications, the risk increasing with the degree of obesity, including hypertensive disorders and preeclampsia, gestational diabetes, and section rates [53]. Moreover, newborns of women with overweight or obesity are more likely to be premature and of high birthweight [54].

The underlying mechanisms of inflammation and fetal growth restriction may be mediated by inflammation and oxidative stress, which are associated with shallow placental invasion and abnormal vascular development, leading to placental blood circulation problems [39,49,55]. This inflammatory and oxidative stress response may be particularly relevant in overweight or obese women before pregnancy, which is aligned with our results.

Our results did not show any correlation between the occurrence of such diseases as chronic hypertension, gestational hypertension, and preeclampsia and DII score during pregnancy. These findings are aligned with the findings described by Sen et al. [24].

### 4.4. Strengths and Limitations

In our study, dietary information came from a validated semiquantitative FFQ, which was designed to evaluate maternal dietary intake in the present study population [29]. Moreover, pregnant women completed dietary assessment (19–23 weeks of gestation) before their oral glucose tolerance test (24–28 weeks of gestation), which means that GDM diagnosis, treatment, or dietary changes could not affect dietary information. Our study also has some limitations. Although the FFQ was validated in the present study population, misclassification of study participants due to measurement errors may have occurred. Furthermore, the average consumption frequency of seasonal foods is especially critical, and the fixed food list in fixed portion sizes is another source of measurement error. Finally, the use of the FFQ to present data on absolute intakes of foods and nutrients is limited without prior calibration of these data by a reference method. No validation of the DII score was performed for the present study population. Our study includes pregnant women at 19–23 weeks’ gestation, and perinatal outcomes may be associated both with the diet before pregnancy and diet during pregnancy. Moreover, we collected dietary information on MD adherence at 19–23 weeks’ gestation and were not able to differentiate between the pre-pregnancy diet and the dietary changes due to pregnancy. However, participants showing low MD adherence at 19–23 weeks’ gestation would probably have the same adherence score and/or dietary habits. In addition, the cross-sectional design of the present study, which does not allow attributing conclusions to plausible causes, and potential residual confounding are limitations of the present study. Additionally, the generalizability of our findings may be limited due to the study participants’ demographic characteristics.

## 5. Conclusions

At mid-gestation, pregnant women with an anti-inflammatory diet profile showed a high adherence to MD. In those women, DII score was associated with pre-pregnancy BMI and the newborn’s birthweight percentile. Nutritional interventions during pregnancy aiming to improve dietary patterns could be an effective measure to improve the maternal dietary inflammatory profile.

## Figures and Tables

**Figure 1 nutrients-14-02284-f001:**
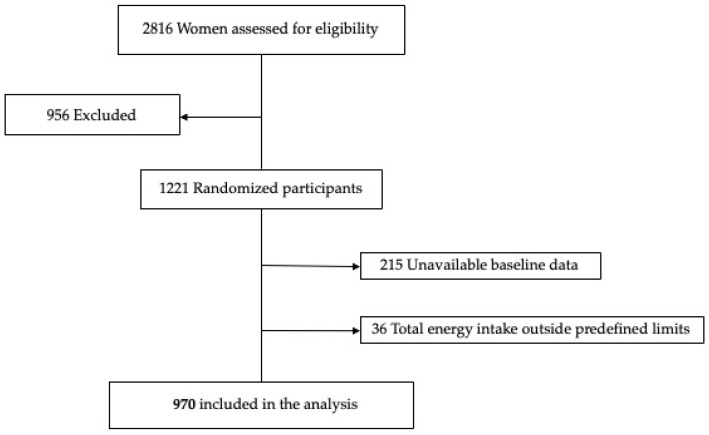
Flow chart of the study population.

**Table 1 nutrients-14-02284-t001:** Baseline characteristics of study participants according to DII score tertiles.

Characteristics	All *n* = 970	DII			*p* ^y^
		Tertile 1 *n* = 323	Tertile 2 *n* = 324	Tertile 3 *n* = 323	
DII *					
Range	−5.71 to −0.33	−5.71 to −3.48	−3.47 to −2.34	−2.33 to −0.33	<0.001
Mean ± SD	−2.94 ± 1.12	−4.23 ± 0.51	−2.89 ± 0.34	−1.7 ± 0.45	<0.001
* **Maternal characteristics** *					
Age at enrollment (years)	37.1 ± 4.70	37.4 ± 4.54	37.2 ± 4.70	36 ± 4.82	0.083
≤24 years	16 (1.6)	4 (1.2)	4 (1.2)	8 (2.5)	
25–29 years	61 (6.3)	18 (5.6)	19 (5.9)	24 (7.4)	
≥30 years	893 (92.1)	301 (93.2)	301 (92.9)	291 (90.1)	
Race and ethnicity ^a^					0.055
Asian	15 (1.5)	6 (1.9)	6 (1.9)	3 (0.9)	
Black	15 (1.5)	5 (1.5)	5 (1.5)	5 (1.5)	
Latin American	128 (13.2)	59 (18.3)	35 (10.8)	34 (10.5)	
White	794 (81.9)	245 (75.9)	271 (83.6)	278 (86.1)	
Maghreb	18 (1.9)	8 (2.5)	7 (2.2)	3 (0.9)	
Socioeconomic status ^b^					0.921
High	618 (63.7)	206 (63.8)	209 (64.5)	203 (62.8)	
Medium	309 (31.9)	105 (32.5)	99 (30.6)	105 (32.5)	
Low	43 (4.4)	12 (3.7)	16 (4.9)	15 (4.6)	
Pre-pregnancy BMI (kg/m^2^)	23.88 ± 4.80	23.4 ± 4.98	23.7 ± 4.28	24.5 ± 5.05	0.011
<18.5 (*n* (%))	61(6.3)	28 (8.7)	14 (4.3)	19 (5.9)	
18.5–24.9	593 (61.1)	201 (62.2)	207 (63.9)	185 (57.3)	
≥25	316 (32.6)	94 (29.1)	103 (31.8)	119 (36.8)	
Systolic blood pressure (mmHg)	105.36 ± 12.02	105.79 ± 13.37	105.48 ± 11.67	104.79 ± 10.85	0.564
Diastolic blood pressure (mmHg)	67.73 ± 8.58	67.3 ± 8.97	68.18 ± 8.37	67.71 ± 8.37	0.424
Education level (schooling years)					0.500
University (yes (%))	653 (67.3)	220 (68.1)	220 (67.9)	213 (65.9)	
Vocational (yes (%))	76 (7.8)	23 (7.1)	29 (9)	24 (7.4)	
Secondary (yes (%))	198 (20.4)	68 (21.1)	59 (18.2)	71 (22)	
Primary (yes (%))	34 (3.5)	7 (2.2)	13 (4)	14 (4.3)	
No education (yes (%))	9 (0.9)	5 (1.5)	3 (0.9)	1 (0.3)	
Use of assisted reproductive technologies (yes (%))	262 (27.0)	82 (25.4)	89 (27.5)	91 (28.2)	0.709
Medical history					
Autoimmune disease (yes (%))	159 (16.4)	43 (13.3)	59 (18.2)	57 (17.6)	0.184
Obesity ^c^ (yes (%))	109 (11.2)	29 (9)	31 (9.6)	49 (15.2)	0.106
Thyroid disorder (yes (%))	110 (11.3)	34 (10.5)	38 (11.7)	38 (11.8)	0.852
Diabetes (yes (%))	48 (4.9)	18 (5.6)	14 (4.3)	16 (5)	0.764
Minor psychiatric disorder ^d^ (yes (%))	38 (3.9)	10 (3.1)	16 (4.9)	12 (3.7)	0.470
Chronic hypertension (yes (%))	41 (4.2)	15 (4.6)	14 (4.3)	12 (3.7)	0.837
Chronic kidney disease (yes (%))	23 (2.4)	6 (1.9)	10 (3.1)	7 (2.2)	0.565
Adverse obstetric history					
Previous SGA (yes (%))	157 (16.2)	44 (13.6)	55 (17)	58 (18)	0.292
Previous preterm birth (yes (%))	54 (5.6)	21 (6.5)	15 (4.6)	18 (5.6)	0.583
Previous preeclampsia (yes (%))	49 (5.1)	17 (5.3)	21 (6.5)	11 (3.4)	0.198
Previous stillbirth (yes (%))	26 (2.7)	6 (1.9)	11 (3.4)	9 (2.8)	0.476
Nulliparous (yes (%))	400 (41.2)	128 (39.6)	146 (45.1)	126 (39)	0.227
During pregnancy					
Smoking habit (yes (%))	64 (6.6)	16 (5)	23 (7.1)	25 (7.7)	0.648
Alcohol consumption (yes (%))	14 (1.4)	6 (1.9)	3 (0.9)	5 (1.5)	0.066
Drug consumption (yes (%))	3 (0.3)	0 (0)	2 (0.6)	1 (0.3)	0.296
Physical exercise (yes (%))	219 (22.6)	98 (30.3)	72 (22.2)	49 (15.2)	<0.001
Yoga or pilates (yes (%))	189 (19.5)	91 (28.2)	64 (19.8)	34 (10.5)	<0.001
Gestational age at randomization, mean (SD), weeks	20.85 ± 0.68	20.85 ± 0.69	20.85 ± 0.68	20.85 ± 0.67	0.999
* **Perinatal outcome** *					
GDM (yes (%))	93 (9.6)	23 (7)	42 (12.6)	28 (9)	0.282
Preeclampsia (yes (%))	72 (7.4)	33 (10.1)	19 (5.7)	20 (6.5)	0.587
Birthweight, mean (SD), g	3195.74 ± 523.4	3246.79 ± 501.73	3185.14 ± 500.5	3155.28 ± 562.91	0.077
Cesarean section (yes (%))	327 (33.7)	104 (31.8)	121 (36.3)	102 (32.9)	0.126
Gestational age at delivery, mean (SD), weeks	39.42 ± 1.76	39.54 ± 1.51	39.4 ± 1.8	39.31 ± 1.94	0.246
Birthweight percentile, mean (SD)	42.66 ± 29.96	44.22 ± 30.56	42.33 ± 28.79	41.43 ± 30.5	0.481
SGA (yes (%))	161 (16.6)	65 (19.9)	46 (13.8)	50 (16.1)	0.400
Severe SGA (yes (%))	60 (6.2)	31 (9.5)	15 (4.5)	14 (4.5)	0.228
Prematurity (yes (%))	60 (6.2)	20 (6.1)	19 (5.7)	21 (6.8)	0.695
Combined adverse perinatal outcome (yes (%))	209 (21.5)	85 (26)	61 (18.3)	63 (20.3)	0.485

DII: Dietary Inflammatory Index. SD, standard deviation; BMI: body mass index; GDM: gestational diabetes mellitus; SGA: small for gestational age. * Continuous variables are presented as mean ± SD, categorical variables are shown as *n* (%). ^y^
*p* values are from ANOVA analysis of variance for continuous data and from χ^2^ tests for categorical data. ^a^ Race and ethnicity were self-reported by the participants. ^b^ Socioeconomic status was defined as low if participants reported having never worked or being unemployed for more than 2 years and having a partner with unqualified work or who was unemployed, high if they reported university studies regardless of whether they were working, and medium in any other situations. ^c^ Obesity was defined as body mass index (BMI; calculated as weight in kilograms divided by height in meters squared) greater than 30. ^d^ Defined as disorders without treatment during pregnancy, either because they were minor disorders or because they were not active during pregnancy.

**Table 2 nutrients-14-02284-t002:** Main food intake according to DII score tertiles (*n* = 970).

Characteristics	All *n* = 97	DII			*p* ^y^
		Tertile 1 *n* = 323	Tertile 2 *n* = 324	Tertile 3 *n* = 323	
EVOO (g/day)	50 (25 to 50)	50 (25 to 50)	50 (25 to 50)	25 (10 to 50)	0.001
Refined olive oil (g/day)	0 (0 to 6.96)	0 (0 to 0)	0 (0 to 9.46)	0 (0 to 10)	0.073
Total nuts (g/day)	12.86 (4.29 to 27.71)	25.71 (12.86 to 42.57)	12.86 (4.29 to 25.71)	6 (0 to 12.86)	<0.001
Vegetables (g/day)	269.15 (198.67 to 355.9)	376.81 (307.07 to 446.77)	269.04 (214.76 to 327.8)	192.78 (141.93 to 234.81)	<0.001
Legumes (g/day)	42.86 (30 to 64.29)	56.19 (41.43 to 79.52)	42.86 (30 to 57.98)	33.33 (20 to 43.33)	<0.001
Fruits (g/day)	306.43 (207.14 to 412.5)	388.57 (303.57 to 513)	307.71 (218.57 to 406.67)	218.86 (148.21 to 307.82)	<0.001
Refined cereals (g/day)	60 (32.14 to 89.71)	51.43 (17.14 to 77.14)	60 (34.14 to 94.29)	72.57 (38.29 to 94.29)	<0.001
Whole grain cereals (g/day)	25.71 (0 to 60)	47.14 (12.79 to 70.61)	31.07 (4.29 to 60)	8.57 (0 to 47.14)	<0.001
Fish or seafood (g/day)	68 (42.6 to 95.9)	83.14 (55.98 to 113.95)	65.9 (42.33 to 92.57)	55.67 (34.39 to 77.87)	<0.001
Blue fish (g/day)	8.33 (0 to 17.86)	17.86 (8.33 to 17.86)	8.33 (0 to 17.86)	8.33 (0 to 17.86)	<0.001
Lean meat (g/day)	74.29 (42.86 to 85.71)	74.29 (52.86 to 85.71)	74.29 (52.86 to 85.71)	69.29 (41.43 to 85.71)	0.106
Processed meat (g/day)	28.57 (14.29 to 50)	24.76 (10.48 to 49.29)	28.57 (21.43 to 50)	28.57 (14.29 to 50)	0.085
Pastries, cakes, or sweets (g/day)	30.1 (13.74 to 55.76)	28.57 (12.42 to 55.71)	30.24 (14.29 to 50.59)	30.48 (12.41 to 60.71)	0.782
Dairy products (g/day)	300 (184.42 to 410.18)	312.5 (174.64 to 440.48)	310.36 (196.61 to 427.32)	275 (187.86 to 366.13)	0.019
Onion (g/day)	15 (5 to 15)	15 (15 to 27.5)	15 (15 to 15)	15 (5 to 15)	<0.001
Garlic (g/day)	0.3 (0 to 0.9)	0.9 (0.3 to 1.65)	0.9 (0.14 to 0.9)	0.14 (0 to 0.3)	<0.001
Oregano (g/day)	2.14 (0 to 2.14)	0.21 (0.07 to 0.21)	0.21 (0 to 0.21)	0.07 (0 to 0.21)	<0.001
Pepper (g/day)	0 (0 to 2.14)	0.33 (0 to 2.14)	0 (0 to 2.14)	0 (0 to 0.71)	<0.001
Alcohol (g/day)	0 (0 to 0.09)	0 (0 to 0.19)	0 (0 to 0.09)	0 (0 to 0.09)	0.002
MD Score	8 (6 to 10)	9 (7 to 11)	8 (6 to 9)	6 (4.25 to 8)	<0.001

DII, dietary inflammatory index; EVOO, extra-virgin olive oil; MD, Mediterranean diet; SD, standard deviation. Data presented are median (25th to 75th percentile) within each DII tertile. ^y^
*p* value refers to the comparison between different tertiles of the DII using Kruskal–Wallis test.

**Table 3 nutrients-14-02284-t003:** Main dietary nutrient intake according to DII score tertiles (*n* = 970).

Characteristics	All *n* = 970	DII			*p* ^y^
		Tertile 1 *n* = 323	Tertile 2 *n* = 324	Tertile 3 *n* = 323	
Energy (kcal/day)	2412.32 ± 449.4	2717.83 ± 361.31	2421.29 ± 365.41	2097.82 ± 387.77	<0.001
Protein (g/day)	102 ± 23.09	115.35 ± 21.25	102.05 ± 19.81	88.6 ± 19.99	<0.001
Carbohydrate (g/day)	211.8 (179.53 to 248.39)	243.56 (213.2 to 278.39)	212.07 (186.52 to 241.83)	181.07 (155.53 to 211.33)	<0.001
Fiber (g/day)	32.04 (25.36 to 39.48)	41.83 (37.53 to 47.19)	32.21 (28.28 to 35.54)	23.64 (19.83 to 26.74)	<0.001
Total fat (g/day)	129.54 (110.8 to 151.81)	144.41 (126.82 to 164.92)	129.81 (113.25 to 150.84)	115.27 (97.8 to 132.27)	<0.001
SFAs (g/day)	34.13 (28.47 to 40.52)	36.73 (31.31 to 42.69)	34.53 (29.03 to 40.47)	30.71 (26.69 to 38.29)	<0.001
MUFAs (g/day)	66.17 ± 17.23	72.91 ± 15.51	66.74 ± 17.28	58.86 ± 15.92	<0.001
PUFAs (g/day)	18.04 (14.75 to 23.1)	22.82 (18.55 to 27.77)	18.2 (15.5 to 22.16)	14.67 (12.91 to 17.48)	<0.001
Linoleic acid (g/day)	13.07 (10.53 to 17.1)	16.55 (12.69 to 20.3)	13.07 (10.66 to 16.64)	10.91 (9.39 to 13.34)	<0.001
α-Linolenic acid (g/day)	1.18 (0.94 to 1.72)	1.7 (1.17 to 2.04)	1.17 (0.98 to 1.67)	0.96 (0.79 to 1.19)	<0.001
EPA (g/day)	0.13 (0.08 to 0.19)	0.17 (0.12 to 0.23)	0.13 (0.08 to 0.18)	0.11 (0.06 to 0.15)	<0.001
DHA (g/day)	0.28 (0.15 to 0.38)	0.34 (0.21 to 0.46)	0.27 (0.15 to 0.37)	0.2 (0.1 to 0.33)	<0.001
Trans-FA (g/day)	1.39 (0.9 to 1.93)	1.33 (0.83 to 1.77)	1.37 (0.85 to 1.9)	1.51 (0.98 to 2.1)	0.006
Cholesterol (mg/day)	329.37 (271.94 to 387.11)	352.56 (290.57 to 419.63)	332.07 (280.83 to 390.09)	301.47 (242.94 to 357.17)	<0.001
Vitamins					
Vitamin A (µg/day)	1232.06 (925.37 to 1605.67)	1647.04 (1353.22 to 1995.93)	1239.6 (1018.95 to 1514.61)	880.25 (696.21 to 1093.78)	<0.001
Vitamin C (mg/day)	241.62 (172.22 to 326.08)	348.1 (278.38 to 423.19)	246.67 (191.3 to 295.85)	159.67 (122.7 to 203.88)	<0.001
Vitamin D (µg/day)	4.56 (3.46 to 5.91)	5.65 (4.3 to 7.16)	4.49 (3.44 to 5.66)	3.83 (2.92 to 5)	<0.001
Vitamin E (mg/day)	17.94 (14.85 to 21.73)	22.62 (19.46 to 26.79)	17.7 (15.68 to 20.11)	14.12 (12.07 to 16.39)	<0.001
Vitamin B1 (mg/day)	1.81 (1.54 to 2.11)	2.12 (1.87 to 2.39)	1.81 (1.59 to 2.04)	1.52 (1.31 to 1.77)	<0.001
Vitamin B2 (mg/day)	2.09 (1.76 to 2.45)	2.42 (2.16 to 2.76)	2.09 (1.81 to 2.38)	1.73 (1.48 to 1.99)	<0.001
Vitamin B3 (mg/day)	23.74 (20.21 to 27.28)	27.74 (24.74 to 30.69)	23.49 (20.81 to 26.05)	20.28 (16.84 to 22.98)	<0.001
Vitamin B6 (mg/day)	2.79 ± 0.67	3.4 ± 0.51	2.76 ± 0.43	2.22 ± 0.45	<0.001
Vitamin B9 (µg/day)	471.52 (376.29 to 581.24)	613.43 (555.07 to 700.19)	470.34 (420.74 to 519.52)	347.85 (298.9 to 393.19)	<0.001
Vitamin B12 (µg/day)	6.33 (4.89 to 8.3)	7.5 (5.8 to 9.56)	7.5 (5.8 to 9.56)	7.5 (5.8 to 9.56)	<0.001
Β-carotene (µg/day)	5481.21 (3853.48 to 7339.81)	7627.1 (6353.83 to 9270.43)	5447.07 (4429.53 to 6880.58)	3561.99 (2744.51 to 4845.43)	<0.001
Minerals					
Zinc (mg/day)	11.92 (10.21 to 14.07)	14.34 (12.25 to 15.85)	12.1 (10.84 to 13.6)	10.06 (8.7 to 11.32)	<0.001
Iron (mg/day)	15.95 (13.51 to 18.59)	19.46 (17.65 to 21.74)	15.94 (14.43 to 17.45)	12.71 (11.27 to 14.04)	<0.001
Magnesium (mg/day)	437.41 (359.96 to 510.06)	545.09 (488.22 to 622.47)	440.92 (394.65 to 477.09)	331.41 (294.23 to 379.52)	<0.001
Selenium (µg/day)	101.32 (84.15 to 121.61)	117.71 (98.4 to 136.98)	101.39 (87.62 to 116.1)	86.98 (72.85 to 103.87)	<0.001

DII, dietary inflammatory index; SD, standard deviation; SFAs, saturated fatty acids; MUFAs, monounsaturated fatty acids; PUFAs, polyunsaturated fatty acids; EPA, eicosapentaenoic acid; DHA, docosahexaenoic acid. ^y^
*p* value refers to the comparison between the DII tertiles using ANOVA or Kruskal–Wallis test as appropriate.

**Table 4 nutrients-14-02284-t004:** Association between DII score (per unit and tertile) and pre-pregnancy BMI and newborn birthweight.

Tertiles of DII Score (1 = Lower DII Score and 3 = Higher DII Score)						
Outcome	*n* = 970		Model 1	Model 2	Model 3	Model 4
** *Pre-pregnancy BMI, kg/m^2^* **						
DII continuous	970	22.43 ± 4.1	0.41 (−0.07 to 0.9)	**0.30 (0.10 to 0.49)**	**0.29 (0.09 to 0.49)**	**0.32 (0.12 to 0.52)**
DII tertile						
Tertile 1	323	23.41 ± 4.98	1 (Ref.)	1 (Ref.)	1 (Ref.)	1 (Ref.)
Tertile 2	324	23.72 ± 4.28	0.45 (−0.77 to 1.67)	0.44 (−0.06 to 0.93)	0.41 (−0.09 to 0.92)	0.42 (−0.08 to 0.92)
Tertile 3	323	24.50 ± 5.05	1.35 (−0.03 to 2.72)	**0.82 (0.26 to 1.39)**	**0.80 (0.24 to 1.37)**	**0.88 (0.31 to 1.45)**
** *Birthweight percentile* **						
DII continuous	970	39.87 ± 29.05	−3.17 (−6.57 to 0.24)	**−3.37 (−6.70 to −0.04)**	**−3.62 (−6.94 to −0.30)**	**−4.05 (−7.42 to −0.68)**
DII tertile						
Tertile 1	323	44.22 ± 30.56	1 (Ref.)	1 (Ref.)	1 (Ref.)	1 (Ref.)
Tertile 2	324	42.33 ± 28.79	−2.51 (−11.20 to 6.18)	−2.52 (−10.99 to 5.96)	−3.12 (−11.59 to 5.36)	−3.86 (−12.36 to 4.65)
Tertile 3	323	41.43 ± 30.5	−7.16 (−16.96 to 2.64)	−8.08 (−17.67 to 1.51)	−8.56 (−18.12 to 0.99)	**−9.84 (−19.57 to −0.12)**

DII, dietary inflammatory index; BMI, body mass index. Model 1: unadjusted; Model 2: adjusted for intervention arm (control/stress reduction/MD), body weight, and gestational age at enrollment; Model 3: model 2 + socioeconomic status (low/medium/high), yoga and pilates practice (hours per week), and assisted reproduction techniques (yes/no). Model 4: model 3 + age, parity (nullipara/multipara), ethnicity/race (Asian/Black/Latin American, White/others), alcohol and smoking habits (yes/no).

## Data Availability

Not applicable.

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
