# Peer review of "Maternal Dietary Inflammatory Index during Pregnancy Is Associated with Perinatal Outcomes: Results from the IMPACT BCN Trial"

_nutrients, 2022, doi:10.3390/nu14112284_

Round 1

Reviewer 1 Report

The manuscript entitled „Maternal Dietary Inflammatory Index during Pregnancy Is Associated with Perinatal Outcomes: Results from the IMPACT BCN Trial” presents interesting issue, but some issues should be corrected.

Major:

The major problem associated with the conducted study results from the fact that the data are obtained from the dietary intervention study. Taking this into account, we observe only the dietary intake during pregnancy, which may not be in agreement with the dietary intake before pregnancy. It is a problem, as perinatal outcomes may be associated both with the diet before pregnancy (general diet which is followed during the whole life) and diet during pregnancy. This problem should be extensively discussed and reflected within the study (in Materials and Methods and Discussion Section).

General:

The manuscript should be prepared according to instructions for authors (e.g. Abstract structured but without headings, Figures needs to have titles, full names of Authors should be presented, etc.)

Abstract:

Brief justification of the study should be presented.

Introduction:

This section needs better organization – in the current version there are whole sections without any references, followed by one sentence with multiple references for one information.

Authors should present comprehensive knowledge – not only based on their previous studies, but various studies of various teams from various countries. In the current version of the manuscript, Authors include to Introduction 3 own references, while one their study is described within whole paragraph with detailed results (lines 69-77).

Materials and Methods:

More details associated with applied tools should be presented – FFQ (e.g. how was it constructed, how were the questions asked, etc.), 7-DR (e.g. which databases were used to calculate dietary intake, how were information about dishes transferred into information about food products, etc.)

It seems that Authors did not verify normality of distribution.

Authors should verify normality of distribution and only for parametric data they should present mean and SD, while for non-parametric they should present median, min and max values.

Authors should use statistical tests based on the distributions observed.

Results:

It seems that Authors did not verify normality of distribution.

Authors should verify normality of distribution and only for parametric data they should present mean and SD, while for non-parametric they should present median, min and max values.

Authors should use statistical tests based on the distributions observed.

Discussion:

Authors should reflect all the limitations of the study (e.g. associated with general limitations of FFQs)

Authors should not extensively describe results, but instead they should focus on the adequate references to discuss the results.

Author Response

Reviewer #1:

The manuscript entitled „Maternal Dietary Inflammatory Index during Pregnancy Is Associated with Perinatal Outcomes: Results from the IMPACT BCN Trial” presents interesting issue, but some issues should be corrected.

Major:

The major problem associated with the conducted study results from the fact that the data are obtained from the dietary intervention study. Taking this into account, we observe only the dietary intake during pregnancy, which may not be in agreement with the dietary intake before pregnancy. It is a problem, as perinatal outcomes may be associated both with the diet before pregnancy (general diet which is followed during the whole life) and diet during pregnancy. This problem should be extensively discussed and reflected within the study (in Materials and Methods and Discussion Section).

Response: Thank you for your comment. We agree that it is a main limitation of the present study. Our cohort include only pregnant women at high risk of SGA, which are selected according to the criteria of the Royal College of Obstetrics and Gynaecology at second trimester ultrasound scan (between 19-23 weeks’ gestation). So, we were not able to include them in our study prior to this time of pregnancy. Moreover, we measured the adherence to Mediterranean diet at 19-23 weeks’ gestation, without prior dietary intervention; it is true that we are not able to differentiate pre-pregnancy diet and the dietary changes due to pregnancy status. However, participants showing low Mediterranean diet adherence at 19-23 weeks’ gestation (mean Mediterranean diet score: 7.7 points), would probably have similar adherence score and/or dietary habits prior to pregnancy.

We added in the discussion section this limitation (line 1178-1186): Our study includes pregnant women at weeks 19-23 weeks’ gestation and perinatal outcomes may be associated both with the diet before pregnancy and diet during pregnancy. Moreover, we collected dietary information of MD adherence at 19-23 weeks’ gestation, not being able to differentiate pre-pregnancy diet and the dietary changes due to pregnancy. However, participants showing low MD adherence at 19-23 weeks’ gestation probably they would have the same adherence score and/or dietary habits.

General:

The manuscript should be prepared according to instructions for authors (e.g. Abstract structured but without headings, Figures needs to have titles, full names of Authors should be presented, etc.)

Response: Thank you for your comments. We modified the abstract, figures and authors names.

Abstract:

Brief justification of the study should be presented.

Response: Thank you for your comment, we modified the abstract (lines 23-24).

Introduction:

This section needs better organization – in the current version there are whole sections without any references, followed by one sentence with multiple references for one information.

Response: We accept the reviewer suggestion, we organized it better adding more references where indicated. Please, see lines 43-180.

Authors should present comprehensive knowledge – not only based on their previous studies, but various studies of various teams from various countries. In the current version of the manuscript, Authors include to Introduction 3 own references, while one their study is described within whole paragraph with detailed results (lines 69-77).

Response: We detailed explained the results of the IMPACT trial because the present cohort study is based on the same participants of the IMPACT study. However, we reduced the information of this study and we add new studies more focused on our aim (line 169-176)

Materials and Methods:

More details associated with applied tools should be presented – FFQ (e.g. how was it constructed, how were the questions asked, etc.), 7-DR (e.g. which databases were used to calculate dietary intake, how were information about dishes transferred into information about food products, etc.)

Response: The FFQ used was validated for the present study population. As suggested, we added additional information about this dietary assessment tool in the methods section (line 483-489)

It seems that Authors did not verify normality of distribution.

Response: Thank you for your observation. As you suggested, we tested for normality and modify the data expression (median [25th to 75th percentile]) and the statistical test. We added this information in the statistical analysis section (line 651-654) and in table 2 and 3 footnotes.

Authors should verify normality of distribution and only for parametric data they should present mean and SD, while for non-parametric they should present median, min and max values.

Response: As we previously described, we modify the results and the tables according to your suggestions. However, we expressed the non-parametric data as median [25th to 75th percentile].

Authors should use statistical tests based on the distributions observed.

Response: We thank for the observation, and we have done it accordingly.

Results:

It seems that Authors did not verify normality of distribution.

Response: We previously answered this question (please, see above and changes in this section in the manuscript).

Authors should verify normality of distribution and only for parametric data they should present mean and SD, while for non-parametric they should present median, min and max values.

Response: We previously answered this question (please, see above and changes in this section in the manuscript).

Authors should use statistical tests based on the distributions observed.

Response: We previously answered this question (please, see above and changes in this section in the manuscript).

Discussion:

Authors should reflect all the limitations of the study (e.g. associated with general limitations of FFQs)

Response: Thank you for comments, we added more limitations in the discussion (lines 1046-1054): “Furthermore, average consumption frequency of seasonal foods is especially critical and the fixed food list in fixed portion sizes are other sources of measurement error. Finally, the use of the FFQ to present data of absolute intakes of foods and nutrients is limited without prior calibration of these data by a reference method. No validation of the DII score was performed for the present study population. Our study includes pregnant women at weeks 19-23 weeks’ gestation and perinatal outcomes may be associated both with the diet before pregnancy and diet during pregnancy. Moreover, we collected dietary information MD adherence at 19-23 weeks’ gestation, not being able to differentiate pre-pregnancy diet and the dietary changes due to pregnancy. However, participants showing low MD adherence at 19-23 weeks’ gestation probably they would have the same adherence score and/or dietary habits. In addition, the cross-sectional design of the present study, which does not allow attributing conclusions to plausible causes, and potential residual confounding are limitations of the present study”.

Authors should not extensively describe results, but instead they should focus on the adequate references to discuss the results.

Response: Thank you so much for your appreciation. We rephrase some parts of the discussion section and we added references linked to the main findings.

Reviewer 2 Report

General comments:

Thank you for the opportunity to review this study. It is an interesting study with quite useful finding, in my opinion. Please see the comments below for your consideration.

  • As far as possible please use active rather than passive language. For example, for line 111 to 114:

All women were asked about their dietary habits in the previous 3 months during pregnancy. Participants were categorized according to the 17-item MD adherence score …

We asked all women about their dietary habits in the previous 3 months during pregnancy. The study categorized the participants according to the 17-item MD adherence score …

  • Be consistent with the presentation. For example, choose pro-inflammatory (line 51) or proinflammatory (line 53, 126) and use the same
  • Be consistent with the way you express units. For example, (line 154), decide if you have a space or not between the number and the unit; 5.8mmol/l or 5.8 mmol/l; similarly, mmHg or mm Hg; 34weeks or 34

Specific comments:

Line

Comment

45

involved – involves

50

trimester – trimesters

69

arrives – drawn

70

which

84

trial

94

-Bases – -Based

106

Please write a proper heading for Figure 1

156

it

171

it

295 - 297

The statement below is unclear or incomplete, would you please rephrase it?

It is well known the anti-inflammatory properties of MD, including improvements in pro-inflammatory mediators, such as adhesion molecules, cytokines, chemokines, among others

300

to – with

311

Ours – Our

311

“according to” – “supporting the”

315

to – with

330

may occur – may have occured

Author Response

Reviewer #2:

Thank you for the opportunity to review this study. It is an interesting study with quite useful finding, in my opinion. Please see the comments below for your consideration.

As far as possible please use active rather than passive language. For example, for line 111 to 114: “All women were asked about their dietary habits in the previous 3 months during pregnancy. Participants were categorized according to the 17-item MD adherence score …

“We asked all women about their dietary habits in the previous 3 months during pregnancy. The study categorized the participants according to the 17-item MD adherence score …”

Response: We agree with the reviewer suggestion, and we changed the manuscript accordingly.

Be consistent with the presentation. For example, choose pro-inflammatory (line 51) or proinflammatory (line 53, 126) and use the same

Response: We accept the suggestion and we changed into “pro-inflammatory” throughout the manuscript.

Be consistent with the way you express units. For example, (line 154), decide if you have a space or not between the number and the unit; 5.8mmol/l or 5.8 mmol/l; similarly, mmHg or mm Hg; 34weeks or 34

Response: We accept the suggestion and we put the space where indicated, whereas we left “mmHg”.

Specific comments:

Line

Comment

45

involved – involves

50

trimester – trimesters

69

arrives – drawn

70

which

84

trial

94

-Bases – -Based

106

Please write a proper heading for Figure 1

156

it

171

it

295 - 297

The statement below is unclear or incomplete, would you please rephrase it?

New paragraph (line 1007-1011): “The anti-inflammatory and immunomodulating effects of MD are well-known and encompass downregulating the expression of leukocyte adhesion molecules and decreasing proinflammatory molecules such as interleukins, chemokines or soluble endothelial adhesion molecules, among others”.

300

to – with

311

Ours – Our

311

“according to” – “supporting the”

315

to – with

330

may occur – may have occured

Response: We thank the reviewer’s suggestions, and we changed the manuscript accordingly.

Round 2

Reviewer 1 Report

The manuscript entitled „Maternal Dietary Inflammatory Index during Pregnancy Is Associated with Perinatal Outcomes: Results from the IMPACT BCN Trial” presents interesting issue, but some issues should be corrected.

Materials and Methods:

More details associated with applied 7-DR should be presented (e.g. which databases were used to calculate dietary intake, how were information about dishes transferred into information about food products, etc.)

Author Response

Reviewer #1:

The manuscript entitled “Maternal Dietary Inflammatory Index during Pregnancy Is Associated with Perinatal Outcomes: Results from the IMPACT BCN Trial” presents interesting issue, but some issues should be corrected.

Materials and Methods:

More details associated with applied 7-DR should be presented (e.g. which databases were used to calculate dietary intake, how were information about dishes transferred into information about food products, etc.)

Response: Thank you for your comment. As you previously mention, we have already validate this FFQ in the present study population. However, as you suggested, we added more information about the FFQ and 7-days food registries. We added the food composition database, detailed information about the structure of the FFQ and we specified that the 7-DR had instruction about portion sizes and household measures. We added all this information in methods section 2.2. Assessment of dietary intake:

Food items were listed under 151 food groups: milk and dairy products, cereals and whole grains, vegetables, legumes, sausages, oils and fats, eggs, meat and fish, fast food, canned products, fruit, nuts, sweets and desserts and others (salt and sugar) and alcoholic and non-alcoholic beverages. Food consumption derived by the FFQ and food records was converted into energy and nutrient intake by the CESNID and Moreiras composition tables using traditional recipes [29, 30].  More detailed FFQ-validation can be found [28]. Participants filled out the 7-days dietary recalls. Detailed instructions about portion sizes and how provide this household measures was also included in the food diary. We asked all women about their dietary habits during pregnancy over the 3 months that precede their enrolment in the study”.